# Microstructure and Mechanical Properties of Dual Scaled NbC/Ti_2_AlC Reinforced Titanium–Aluminum Composite

**DOI:** 10.3390/ma16134661

**Published:** 2023-06-28

**Authors:** Sen Cui, Chunxiang Cui, Xin Wang

**Affiliations:** 1School of Materials Science and Engineering, Shijiazhuang Tiedao University, Shijiazhuang 050043, China; 2School of Materials Science and Engineering, Hebei University of Technology, Tianjin 300401, China; hutcui@hebut.edu.cn (C.C.); ahaxin@hebut.edu.cn (X.W.)

**Keywords:** strengthening, TiAl, metal matrix composite

## Abstract

A TiAl composite containing hybrid particles and whisker reinforcements is fabricated by vacuum melting. The results of this study show that the comprehensive mechanical properties and refining effect of the material are best when the content of reinforcement is 1 wt.%, and then the mechanical properties begin to deteriorate as the content increases further. Finely dispersed NbC particles and uniformly dispersed Ti_2_AlC whiskers are the ideal second phases. The synergistic strengthening effect of NbC particles and in situ Ti_2_AlC whiskers are key to the improvement of mechanical properties. Compared with the TiAlNb matrix, the fracture stress/strain of the composite at 1073 K is improved from 612 MPa/19.4% to 836 MPa/26.6%; the fracture toughness at room temperature is improved from 18.8 MPa/m^2^ to 27.4 MPa/m^2^.

## 1. Introduction

Titanium–Aluminum (TiAl) has attracted the attention of scholars due to its high specific strength and good high-temperature performance [1,2,3,4]. The urgent problem to be solved for the TiAl alloy is to improve its brittleness at room temperature and enhance the high-temperature strength. By adding β stabilizing elements such as Nb, Mo and Cr, the phase composition can be adjusted and the degree of segregation of TiAl alloy can be reduced, which has been a widely used method [5,6,7]. On this basis, a composite method is an efficient way to further improve the mechanical performance of cast TiAl alloys. By introducing ceramic phases into the alloy matrix, the comprehensive mechanical properties of metal matrix composites (MMCs) could be significantly improved when comparing with the matrix [8,9]. This is due to the intrinsic mechanical properties of these ceramic reinforcements and their refining effects on the microstructure of the alloy matrix. In the past few decades, different scaled ceramic phases, such as particles [10] and whiskers [11], have both been extensively studied as reinforcements in MMCs. In terms of the strengthening mechanisms, the emphases of particles and whiskers are different. Particle reinforcements have a relatively small size. Therefore, particles with the same volume fraction can provide more heterogeneous nucleation cores than whiskers, and thus have a better refining effect. Moreover, small particles have a greater tendency to distribute inside the grains, which can play a better role in Orowan strengthening in the process of deformation. Whiskers are relatively large in size and can bear the load effectively. Hence, whiskers can play a larger role in the load transferring mechanism. Meanwhile, the bridging effect of whiskers between multiple grains can also delay the fracture of the alloy and improve the comprehensive mechanical properties. 

Based on the above background, both particle reinforcements and whisker reinforcements have specific advantages. It is of great research value to study TiAl matrix composites containing the hybrid particle and whisker reinforcements. 

NbC possesses high strength and a high melting point, and is widely used in alloys to enhance their high temperature strength and abrasion performance [12,13]. Hence, NbC can be the particle reinforcement in TiAl alloys. In the casting process, NbC particles can act as the heterogeneous nucleation cores to refine and strengthen TiAl matrix. MAX phase is a kind of reinforcement with excellent high temperature mechanical properties [14,15,16]. Meanwhile, the generation and growth of Ti_2_AlC/Ti_2_AlN in TiAl alloys have already been proven [17,18,19,20]. MAX phases such as Ti_2_AlC and Ti_2_AlN have good mechanical properties at the service temperature of TiAl alloys, including their high temperature strength and high temperature ductility. Hence, they are ideal whisker reinforcements in TiAl alloys.

This work prepares a TiAl composite containing hybrid particle&whisker reinforcements by casting strategy. NbC is the ex situ particle reinforcement and graphene is used as the reaction source of the Ti_2_AlC phase, which can form in the TiAl melt and grow into whiskers. By controlling the content of NbC and graphene, the second phases of ideal morphology can generate in the TiAl composite, and the synergistic effect of multiplying strengthening mechanisms brought by the hybrid reinforcements helps to improve the mechanical properties of TiAl matrix. 

## 2. Materials and Methods

### 2.1. Preparation of TiAl Composite 

Ti-48Al-6Nb (TiAlNb alloy) is used as the matrix of the TiAl composite, which is fabricated by a vacuum melting furnace. Titanium (99.9%) bulks, Aluminum (99.9%) bulks and commercial Aluminum–Niobium (Al-70 wt.%Nb) intermediate alloy are used as the raw materials, which are prepared by vacuum melting. NbC particles and graphene (mass ratio 6:1) mixed powders are used as the raw materials of reinforcements. NbC particles are about 2 μm in size; flake-like graphene is about 10 μm in width. The mixed powders are ball-milled for 20 h by a GN-2 high energy ball-milling machine; the milled powders are less than 1 μm in size. Then, the mixed powders and TiAlNb alloy are used to prepare the TiAl composite by the vacuum melting furnace. TiAlNb alloy (cuboid, about 100 g in weight) is placed in the crucible and the mixed powders are hung above the crucible. The mixed powders are placed into the TiAl melt after the TiAlNb is fully melted. A high purity Mo stirrer is used to stir the melt containing the mixed powder. The melt is then poured into a copper mold to obtain the final NbC/Ti_2_AlC reinforced TiAl composite (NTTC, φ20 in size, about 100 g in weight, as shown in Appendix A).

### 2.2. Structural Characterization

A Zeiss metalloscope (AxioCam ICc5, Jena, Germany) is used for the observation of the metallographic structure of the TiAlNb matrix and TiAl composite. The microstructure of the different specimens is observed by a scanning electron microscope (SEM, S-4800, Hitachi Ltd., Tokyo, Japan) with EDAX energy-dispersive spectrometry (EDS). Ion beam thinning is used for the preparation of the electron transparent foils of the TiAl composite. High-resolution transmission electron microscopy (HRTEM) images and the selected area electron diffraction (SAED) patterns of the specimens are obtained by JEM-2100 transmission electron microscopy (TEM). 

### 2.3. Mechanical Properties Test

Room-temperature (RT) compression tests are carried out by an SHT5305 universal tester (MTS Ltd, Shanghai, China) with electronic extensometer. The strain rate of compression is ε′= 2.5 × 10^−4^ s^−1^. High-temperature (HT) compression tests are carried out by a Gleeble 3180 (DSi Ltd., Minneapolis, MN, USA) thermal mechanical simulator at 1073 K (a typical service temperature of TiAl alloy) with a strain rate of 0.01 s^−1^. The specimens of RT compression and HT compression tests are φ2 × 4 mm in size and φ8 × 12 mm in size, respectively. A microhardness test was carried out by an HMV-2T microhardness tester (SHIMADZU Ltd., Kyoto, Japan) with a diamond indenter under 980 mN load for 10 s. Each specimen was tested 10 times and the average values were regarded as the results. All the specimens are obtained by wire-electrode cutting followed by mechanical polish. Three-point bending tests are carried out by the SHT5305 universal tester at room temperature. The specimens are 2 × 4 × 20 mm in size with a span length of 16 mm and are all cut by electrical discharge machining followed by mechanical polishing. The bending samples are processed with a slit notch (2 mm in depth and 0.3 mm in width) to calculate the fracture toughness. The fracture toughness is calculated by the following equation [21]: K_IC_ = 3PL(10a)^1/2^/(200bh^2^)[1.93 − 3.07(a/h) + 14.53(a/h)^2^ − 25.07(a/h)^3^ + 25.80(a/h)^4^] (1)
where K_IC_ is the fracture toughness and a is the depth of the slit notch (mm). P is the maximum load (N), L is the span length (mm) and b and h refer to the width (mm) and thickness of the specimens (mm), respectively. The compression speed of cross head is 0.1 mm/min. 

## 3. Results

The morphology and size of NbC/graphene mixed powders are determined by SEM. As shown in Figure 1, it is easy to distinguish the graphene flake and the NbC particles in the mixed powders in Figure 1a. Figure 1b,c show the EDS results of NbC and graphene, respectively. The flakes of graphene are about 10 μm in width. The spherical particles of NbC are about 2 μm in size. The flake-like graphene has a large size and floats on top of the NbC particles. Hence, this mixed powder needs to be processed by ball-milling to refine the size and reduce the segregation of the graphene. 

Figure 2 shows the morphology of the mixed powders after ball-milling. Figure 2a,c,e show the 500× micrographs of powders after 6 h, 12 h and 20 h of ball-milling. Figure 2b,d,e show the corresponding images of the microstructure at 5000×, respectively. The evolution of powder morphology at 500× shows that the size of the graphene flakes decreases gradually. Meanwhile, the 5000× micrographs show that the dispersion degree increases as the ball-milling time increases. The morphology evolution from Figure 2b,d,f shows that the uniformity of the mixed powders is improved significantly; meanwhile, the size of the powders is decreased significantly.

The metallographic microstructures of the TiAlNb alloy and NTTC are exhibited in Figure 3. The grain size of the TiAlNb alloy is about 400 μm, as shown in Figure 3a. Moreover, micrometer-scale lamellas can be observed inside the grains. NTTC samples with different mass fraction (0.5%, 1%, 2%, 3%) are named as T0.5, T1, T2 and T3, respectively. The grain size of the TiAlNb matrix is about 400μm. With the addition of NbC and graphene, T0.5 in Figure 3b shows a much-refined grain size of nearly 100 μm. As the powder amount increases to 1%, the grain size of the T1 sample (Figure 3c) is slightly decreased compared with T0.5. However, as the mass fraction of NbC–graphene powders increases further, the grain size of the T2 sample in Figure 3d shows no significant difference with T1. Figure 3e shows the microstructure of the T3 sample, which differs greatly from those of the other samples. The content of the second phase increases significantly, and its grain size cannot be determined. In the high magnification image in Figure 3e, the segregation of the second phase could be observed. 

The SEM images of NTTC are shown in Figure 4. The microstructure of T0.5, T1, T2 and T3 is displayed in Figure 4a–d, respectively. The rod-like phases in T0.5 and T1 show uniform distribution. As the mass fraction reaches 2%, the content of the second phases in T2 increases obviously; meanwhile, some particles tend to gather, which denotes that the segregation in the T2 sample has already begun. After this, the segregation of the second phase becomes more obvious in the T3 samples, as shown in Figure 3f; the second phase loses its rod-like appearance and irregular bulk appears.

For qualitative analysis of the rod-like second phase, the T1 samples were selected for further SEM and EBSD analysis, as shown in Figure 5. Figure 5a shows the SEM images showing the distribution of the rod-like second phases. The relative EBSD image of Figure 5a is shown in Figure 5b. The EBSD result shows that each rod has a single crystallographic orientation, which denotes that the rod-like second phases are in situ whiskers.

To further determine the phase formation of the whiskers and observe the distribution of ex situ NbC particles, the microstructure of the T1 sample inside the grains is analyzed, as shown in Figure 6. Figure 6a exhibits that the whiskers can be located both inside the grain and in the grain boundary. The insert is the EDS result of Point 1 in Figure 6a. According to this result, the atomic ratio of element Ti, Al and N is nearly 2:1:1; hence, the whiskers can be identified as Ti_2_AlC phase. Figure 6b shows the NbC particles which distribute in the lamellas inside the grain. Meanwhile, the refined lamellas can be observed in Figure 6, which shows roughly dozens of nanometers. 

Figure 7 shows the TEM images of the T1 sample. Figure 7a shows the α_2_/γ lamellas in T1, which are less than 100 nm. Compared with the micrometer scale lamellas in the TiAlNb matrix, the lamellas in T1 have been significantly refined. Figure 7b shows the SAED patterns of the diaphasic lamellas in Figure 7a; the calibrate results show the typical Blackburn orientation relationship, which is the parallel relation between (111)_γ_ and (0001)_α2_. Figure 7c shows a Ti_2_AlC whisker located in the lamellas. The HRTEM image of the interface between this whisker and the TiAl matrix is shown in Figure 7d, in which the parallel relationship between the two phases can be observed. The FFT patterns of Figure 7d are exhibited in Figure 7e. According to the calibration results, the specific orientation relationship between the two phases can be determined, which is (111)_γ_//(0006)_Ti2AlC_.

In order to analyze the mechanical properties of the NTTC and TiAlNb matrix, different tests are carried out, as shown in Figure 8. Figure 8a shows the room-temperature compression curves of the different samples. The fracture stress and strain of TiAlNb, T0.5, T1, T2 and T3 are 1453 MPa/17.8%, 1887 MPa/22.8%, 2247 MPa/24.4%, 2183 MPa/22.0% and 2063 MPa/18.6%, respectively. Figure 8b shows the compression curves of the different samples at 1073 K, the fracture stress and strain of TiAlNb, T0.5, T1, T2 and T3 are 612 MPa/19.4%, 769 MPa/24.9%, 836 MPa/26.6%, 798 MPa/20.2% and 753 MPa/16.7%, respectively. The fracture toughness of TiAlNb, T0.5, T1, T2 and T3 are 18.8 MPa/m^2^, 25.0 MPa/m^2^, 27.4 MPa/m^2^, 25.6 MPa/m^2^ and 21.1 MPa/m^2^, respectively, which are exhibited in Figure 8c. All the values are also presented in Table 1. (The Parallel test results of room temperature compression tests are shown in Appendix A) The mechanical properties of samples show that, with the addition of reinforcements, the mechanical properties of the TiAl alloy firstly improve and then deteriorate. T1 shows the best comprehensive mechanical properties. The microhardness of the different samples is shown in Table 2. The microhardness increases as the amount of reinforcements increases. 

## 4. Discussion

### 4.1. The Refining Effects of Dual Reinforcements

Compared with the TiAlNb matrix, the grain size of the T1 sample was refined from 400 μm to less than 100 μm, and the lamellar spacing was refined from a few microns to less than 100 nm. The refining is attributed to the second phases, which act as the heterogeneous nuclei. The schematic diagrams are shown in Figure 9, which exhibits the evolution process of the second phases. Ex situ NbC has a melting point of 3500 °C, which can distribute stably in TiAlNb melt (Figure 9a). After that, as graphene reacts with Ti, TiC begins to generate in the melt (Figure 9b). Then, during the cooling process, TiC tends to react to transform into Ti_2_AlC, which it does according to the transformation pathway L + TiC → L + Ti_2_AlC [22]. Meanwhile, the Ti_2_AlC can spontaneously grow into whiskers in the melt, as shown in Figure 9c. During the subsequent cooling process, NbC particles and Ti_2_AlC whiskers will act as nucleation substrates and TiAlNb nucleates grow on the two second phases (Figure 9d). Abundant NbC particles, along with the Ti_2_AlC whiskers, provide plenty of nucleation sites for the nucleation of the TiAl melt, which refines the grain size and lamellar width significantly. Due to the different sizes of the two reinforcements, the number of NbC particles is much more than that of Ti_2_AlC whiskers. Therefore, NbC particles play a leading role in the refining effect of TiAlNb. 

### 4.2. The Synergistic Strengthening Mechanisms 

The increase in the strength of NTTC is attributed to the synergistic strengthening mechanisms brought by the hybrid NbC particles and Ti_2_AlC whiskers, including fine grain strengthening, Orowan strengthening, Load Transferring (LT) strengthening and Coefficient of thermal expansion (CTE) mismatch strengthening. 

The mechanical properties of the TiAl alloy are significantly influenced by the grain size and lamellar width. As the lamellar width is refined, the mechanical properties of TiAl are improved, such as the yield strength, ductility, fracture toughness and creep resistance. According to the Hall–Petch equation, yield strength σy of the alloy can be described as [23]:(2)$$\sigma_{y}=\sigma_{0}+\frac{K_{y}}{\sqrt{d}}$$
where σ_0_ is the lattice frictional resistance due to single dislocation movement, K_y_ is a material constant and d is the average grain size. As the grain size and lamellar width are both refined, the number of grain boundaries [24] and lamellar interfaces increases. As a result, the resistance to dislocation motion is greatly enhanced, thus increasing the strength of the TiAlNb alloy. According to the analysis of the refining effect, in the two reinforcements, NbC plays the main role in the fine grain strengthening mechanism.

When the dislocation moves near the second phase, due to the high strength of the second phase, the dislocation is unable to pass through the particle but bypasses them, leaving a dislocation loop around the particle, which is called the pinning effect on dislocation [25,26]. These bending dislocation lines will increase the lattice distortion energy in the affected region of dislocation and increase the resistance of the dislocation movement, thus resulting in an increase in material strength. The effect of the Orowan strengthening mechanism can be described by the following equation [27]:(3)$$\sigma_{\mathrm{Orowan}}=\frac{\mathrm{MGb}}{d_{P}}{\left(\frac{{6V}_{R}}{\pi }\right)}^{1/3}$$
where M denotes the Taylor factor, G is the shearing modulus of TiAlNb matrix, b is the burger vector, d_p_ is the diameter of the second phase particles and V_R_ is the volume fraction of the second phases. It is easy to notice that for the second phase with a given volume fraction, the smaller the diameter, the stronger the effect of Orowan strengthening. 

Ti_2_AlC whiskers have a much larger scale than the NbC particles; hence, the Ti_2_AlC/TiAl interfacial stability has a much greater influence on the mechanical properties of the composite, which should be analyzed. According to the HRTEM image and FFT patterns in Figure 7d,e, the parallel relationship between (0006)_Ti_2_AlC_ and (111)_γ−TiAl_ has already been described. The planar mismatch value δ of (0006)_Ti2AlC_/(111)_γ−TiAl_ is calculated to be 2.59% (d(0006)_Ti2AlC_ = 0.226 nm, d(111)_γ−TiAl_ = 0.232 nm), which is smaller than 5%. Hence, the bonding mode of this interface belongs to the coherent interface. The coherent interface can effectively promote the transfer of load between the matrix and whisker. Thus, the excellent mechanical properties of whisker reinforcement can be fully utilized. In the service process, whiskers can effectively share the external load through the coherent interface so as to improve the comprehensive mechanical properties of materials and delay the fracture and failure of the materials.

Due to the difference in thermal expansion coefficient α between the matrix and the reinforcements (α_TiAl_ = 10.4~11.1 × 10^−6^ K^−1^, α_NbC_ = 6.65 × 10^−6^ K^−1^, α_Ti2AlC_ = 8.2 × 10^−6^ K^−1^), the microscopic deformation of the matrix and reinforcements is different in the process of solidification, resulting in the arising of geometrically necessary dislocations and leading to the enhancement of material strength, which is named as the coefficient of thermal expansion (CTE) strengthening [28,29]. The contribution of CTE mismatch to the strength can be determined by [30]:(4)$$\sigma_{\mathrm{CTE}}^{}=\sqrt{3}\beta G\sqrt{\rho_{\mathrm{CTE}}}$$
(5)$$\rho_{\mathrm{CTE}}=\frac{12V_{R}\Delta \alpha \Delta T}{1-V_{R}-{\mathrm{bd}}_{R}}$$
where β is the dislocation strengthening coefficient and ρ_CTE_ is the dislocation density, which is caused by the difference in thermal expansion coefficient between the matrix and the reinforcements. d_R_ is the particle size of the reinforcement. Δα is the CTE difference between the matrix and the nanoparticles and ΔT is the temperature change. Due to the existence of the mismatch of CTE, both NbC and Ti_2_AlC contribute to the CTE strengthening. 

The toughening of NTTC is mainly attributed to the grain refining effect and the effect of whiskers. By bearing the load which is transferred from the matrix, the fracture of the composite can be delayed. Meanwhile, when cracks start to extend, whiskers can cause the deflection of these cracks, thus increasing the distance of crack propagation and delaying the fracture.

In accordance with the microstructure and mechanical properties shown in Figure 3, Figure 4 and Figure 8, the amount of reinforcements influences the morphology and mechanical properties of MMCs. Especially when the content of reinforcements reaches a certain value, the ceramic phases will aggregate and deteriorate the properties. The aggregated bulk easily causes stress concentration, which causes the material to fracture under a small strain. Therefore, the total energy absorbed by the material from deformation to fracture is also reduced, which is reflected in the reduction of fracture toughness of the T3 sample compared with the T2 sample. This is a common phenomenon in MMCs containing particle reinforcement. Under the condition of the same volume fraction, whisker second phase has obvious excellent advantages compared with the agglomeration particles’ second phase. Therefore, the dual scaled reinforcements in this study have the advantage of adjusting the morphology of second phases. 

## 5. Conclusions

NbC particles and in situ Ti_2_AlC whiskers are introduced into a TiAl alloy by a casting method; the mechanical properties of this composite are influenced by the content of the reinforcements. The key to this method is to control the morphology and distribution of the reinforcements. NTTC with 1 wt.% NbC/graphene powders exhibits the best mechanical properties, in which the second phases have ideal morphology and distribution. Particle reinforcements alone with the in situ Ti_2_AlC whiskers can improve the comprehensive mechanical properties of the TiAl matrix by the synergistic strengthening mechanisms. The future development direction of TiAl composites in this work is mainly to reduce its casting defects by hot isostatic pressing (HIP) treatment to obtain high-temperature structural materials with low cost and high performance.

## Figures and Tables

**Figure 1 materials-16-04661-f001:**
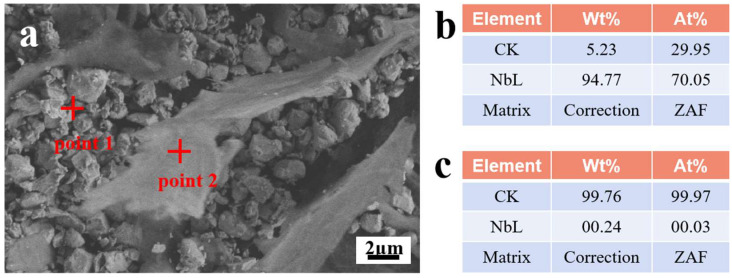
SEM image and EDS patterns of mixed powders. (**a**) SEM image showing the morphology of NbC and graphene. (**b**) and (**c**) are the EDS results of point 1 and point 2 in Figure 1a, respectively.

**Figure 2 materials-16-04661-f002:**
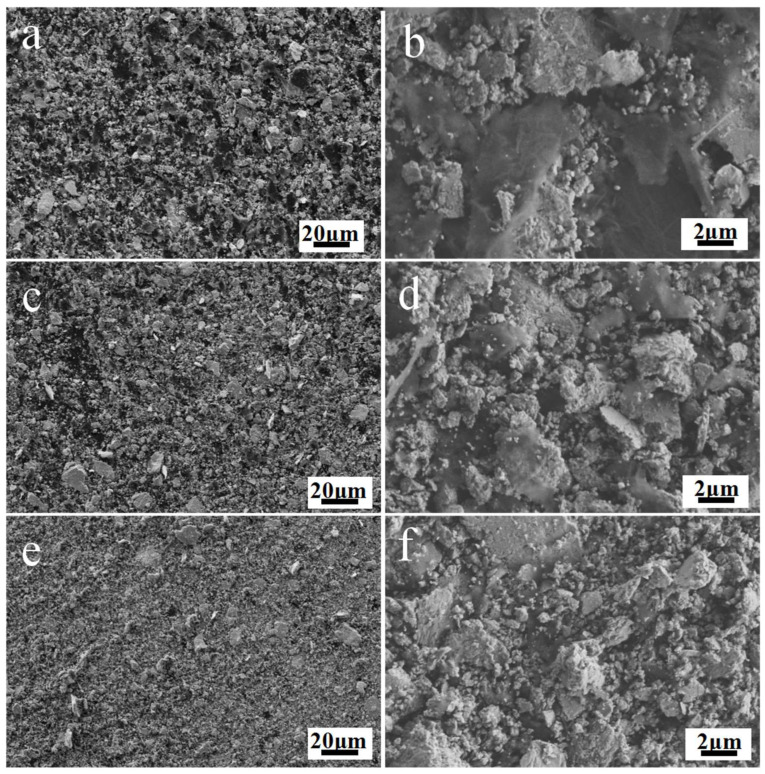
SEM images showing the NbC–graphene mixed powders after different ball-milling times. (**a**,**c**,**e**) are the microstructure of NbC–graphene after 6 h, 12 h and 20 h ball-milling at 500×, respectively. (**b**,**d**,**f**) are the micrographs at 5000× of powders in Figure 2a,c,e, respectively.

**Figure 3 materials-16-04661-f003:**
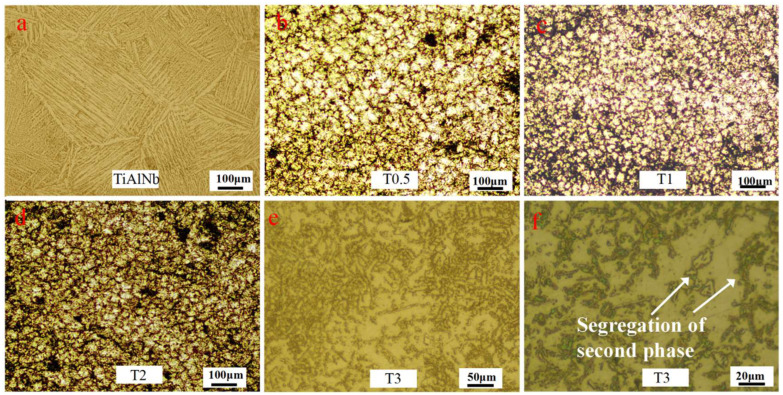
Microstructure of TiAlNb and NTTC. (**a**) Coarse TiAlNb grains. (**b**,**c**,**d**,**e**) are the microstructure of NTTC with 0.5 wt.%, 1 wt.%, 2 wt.% and 3 wt.% NbC–graphene powders (named as T0.5, T1, T2 and T3, respectively). (**f**) refers to the high magnification images of the T3 sample.

**Figure 4 materials-16-04661-f004:**
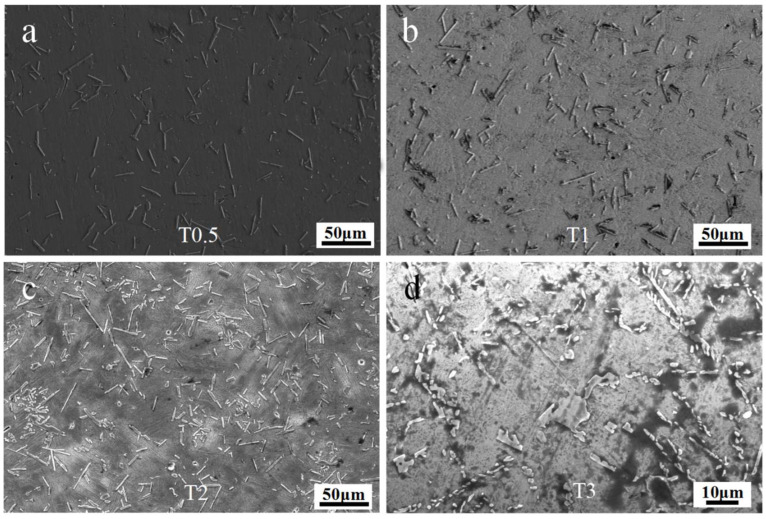
SEM images showing the microstructure of NTTC. (**a**) T0.5. (**b**) T1. (**c**) T2. (**d**) T3.

**Figure 5 materials-16-04661-f005:**
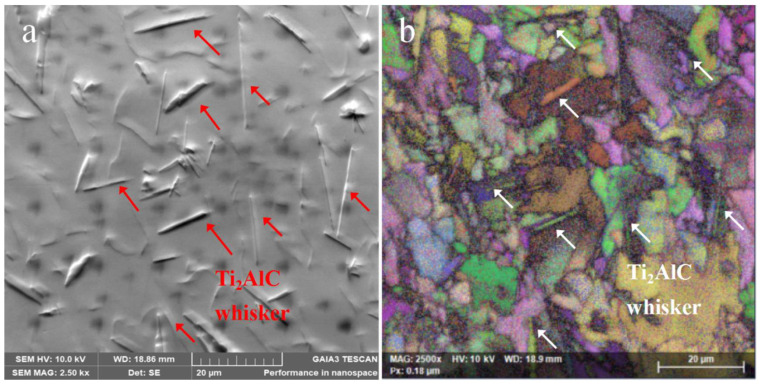
SEM image and the relative EBSD results of T1 sample. (**a**) Microstructure with abundant rod-like second phase. (**b**) EBSD image of (**a**).

**Figure 6 materials-16-04661-f006:**
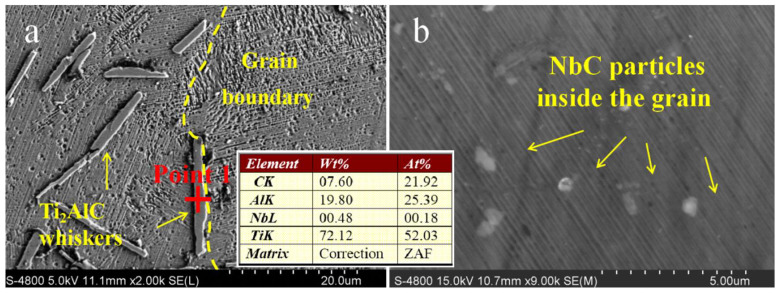
Microstructure of T1 samples. (**a**) Ti_2_AlC whiskers inside the grain and in the grain boundary. (**b**) NbC particles in the lamellas.

**Figure 7 materials-16-04661-f007:**
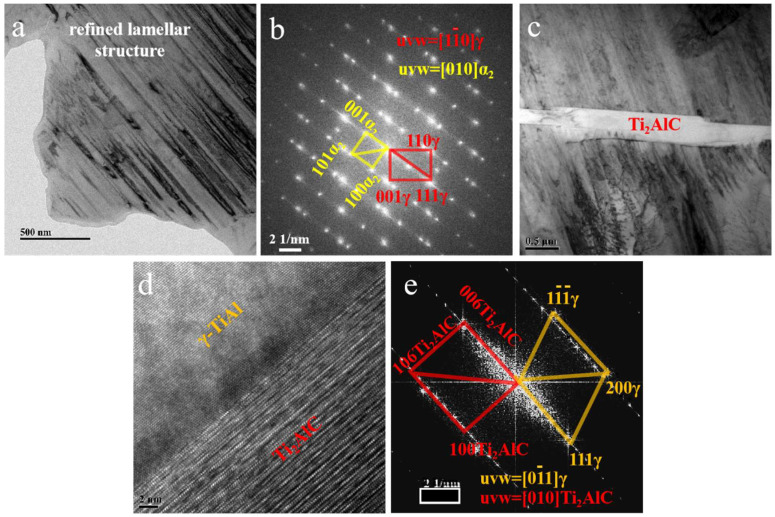
TEM image of T1 sample. (**a**) Refined α_2_/γ lamellar structure in T1 sample. (**b**) SAED patterns of α_2_/γ in (**a**). (**c**) TEM image showing the location of Ti_2_AlC whisker in the lamellas. (**d**) HRTEM image showing the interface between Ti_2_AlC and γ-TiAl. (**e**) FFT patterns of (**d**).

**Figure 8 materials-16-04661-f008:**
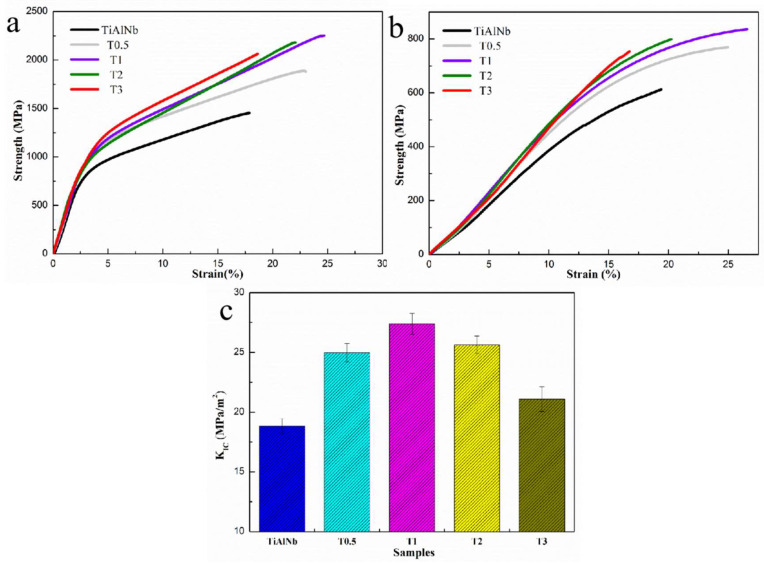
Comparison of mechanical properties between TiAlNb, T0.5, T1, T2 and T3. (**a**) Room-temperature compression curves. (**b**) High-temperature (1073 K) compression curves. (**c**) Fracture toughness results of different samples.

**Figure 9 materials-16-04661-f009:**
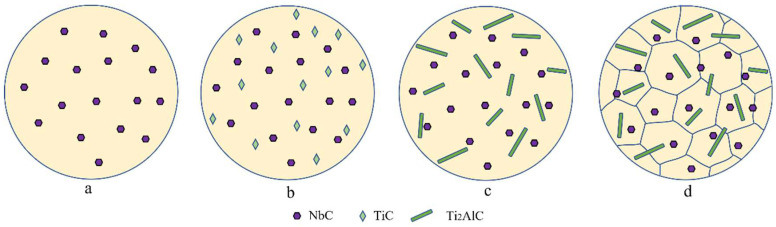
Schematic diagrams showing the formation of Ti_2_AlC whiskers in TiAlNb. (**a**) NbC in TiAlNb melt. (**b**) TiC generates first in TiAlNb. (**c**) Ti_2_AlC generates in TiAlNb. (**d**) TiAlNb nucleates and solidifies.

**Table 1 materials-16-04661-t001:** Mechanical properties of different samples.

	Samples	TiAlNb	T0.5	T1	T2	T3
	Properties					
Room Temperature	Compression Strain (%)	17.8	22.8	24.4	22.0	18.6
	Compression Stress (MPa)	1453	1887	2247	2183	2063
	Fracture toughness (MPa/m^2^)	18.8	25.0	27.4	25.6	21.1
High Temperature (1073 K)	Compression Strain (%)	19.4	24.9	26.6	20.2	16.7
	Compression Stress (MPa)	612	769	836	798	753

**Table 2 materials-16-04661-t002:** Microhardness of different samples.

Samples	TiAlNb	T0.5	T1	T2	T3
Microhardness (HV)	489	521	531	558	567

## Data Availability

All data included in this study are available upon request by contact with the corresponding author.

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
