# Peer review of "Microstructure and Mechanical Properties of Dual Scaled NbC/Ti2AlC Reinforced Titanium–Aluminum Composite"

_materials, 2023, doi:10.3390/ma16134661_

Round 1

Reviewer 1 Report

An original and interesting article is presented on the effect of NbC and Ti2AlC whiskers on the mechanical characteristics of a TiAl composite. I recommend making additions and corrections according to the comments made, which will make the article more understandable for the reader.

1. The introduction begins with the words that materials and alloys based on TiAl are attractive. However, references are made to articles from 2015 and 2016. Reference should be made to more recent sources. For example: https://doi.org/10.1016/j.jallcom.2022.167611, https://doi.org/10.3390/met13061002, https://doi.org/10.1016/j.pnsc.2023.05.002

2. P.1, L. 18-19: as written, it is understood that with increasing temperature, the strength of the material increases, but this is not so.

3. P.2, L. 66-69: it is necessary to indicate the fineness of the powders before and after grinding, and also by what method the Aluminum-Niobium (Al-70 wt%Nb) intermediate alloy was obtained

4. P.2, L. 74-75: it is necessary to indicate the dimensions of the samples received

5. P.2, L.87-88: it is necessary to explain why the compression tests are carried out at a temperature of 1073 K.

6. In Fig. 1 in the tables, the designation of the elements “CK” and “NbL” is not clear

7. P.3, L.106: it is necessary to clarify what is meant by “poor dispersion”

8. P.5, L.175-178: correct figure title

9. During the mixing of powders, they can react with each other to form new compounds. It is necessary to present the XPA results of the powders before and after mixing in order to clarify this point.

10. To comment on the results of fig. 4 on the increase in the amount of phase 2, it is necessary to present the EDA results for each sample.

11. P.4, L.160-168: present the obtained values in the form of a table, the text is not clear and difficult to read

12. P.10, L.261-262: give values of coefficients of thermal expansion for matrix and reinforcements

13. It is well described why mechanical characteristics increase. It is necessary to add a description of why K1C decreases for T2 and T3.

Reviewer 2 Report

Authors reported the microstructure and mechanical properties of dual scaled NbC/Ti2AlC reinforced TiAl composite. This paper needs revision as per the below comments.

(1)                    The full form of abbrevations should be given when it is introduced. Example what is TiAl? And etc.

(2)                     The synergistic strengthening effect of NbC particles and in-situ Ti2AlC whiskers are the key in the improvement of mechanical properties, which is combined by fine grain strengthening, Orowan strengthening, Load Transferring (LT)  strengthening and Coefficient of thermal expansion (CTE) mismatch strengthening - Why strengthening mechanisms have been explained in abstract?

(3)                    The introduction section needs major improvement with few more literatures. Address the research gap properly.

(4)                    NbC particles and graphene (mass ratio 6:1) mixed powders are used as the raw materials of reinforcements. Mixed powders are ball-milling for 20h by a GN-2 high energy ball milling machine – justify the reasons that why these parameters have been used?

(5)                    Room-temperature (RT) compression tests are carried out by SHT5305 universal tester with electronic extensometer – what is the ASTM standard followed?

(6)                    What is given in Figure 2? Figure 2. This is a figure. Schemes follow another format. If there are multiple panels, they should be listed as: (a) Description of what is contained in the first panel; (b) Description of what is contained in the second panel. Figures should be placed in the main text near to the first time they are cited.

(7)                    In mechanical properties, why compressive strength alone tested?

(8)                    Please include the density and hardness of the proposed composites and its variations.

(9)                    Provide the compression test other results as in table format.

(10)                Provide the photo images of the samples produced.

(11)                What would be application of the proposed composites?

(12)                Revise the conclusion with future scope of the work

(13)                Compared with TiAlNb matrix, the grain size of T1 sample was refined from 400 μm to less than 100 μm, and the lamellar spacing was refined from a few microns to less than 100 nm – how the grain size was measured?

(14)                Avoid abbreviations in the title of the paper.

Reviewer 3 Report

Corrections

Line 69

Modify the sentence: Mixed powders are ball-milling for 20h by a GN-2 high energy ball- 69 milling machine

Line 134, 137, 139, 186

Distribution of the rob-like second phases (rod like?)

Line 175-178

Modify the title of Figure 2

Line 201-202

Modify the title of Figure 9

Line 235-236

Modify the text as no continuity

Line 245-246

Denote the parameter dp, VR (the volume fraction) appropriately

Line 268

Define the term dR in equation 5

Suggestions

Discuss about the processing method (casting strategy) vacuum melting?

Discuss how the dispersity is ensured in composite development.

Describe why TiAl composite and its strengthening by MAX phase is important

Specify the testing standard used for the measurement of fracture toughness

Check for the continuity of sentences

Follow the standard way of presenting the units

Round 2

Reviewer 1 Report

All remarks were eliminated and appropriate explanations were given. I recommend accepting the article.

Reviewer 2 Report

The revised paper entitled “Microstructure and mechanical properties of dual scaled NbC/Ti2AlC reinforced Titanium-Aluminum composite” is recommended for publication.